# Expression and Prognostic Implication of PD-L1 in Patients with Urothelial Carcinoma with Variant Histology (Squamous Differentiation or Micropapillary) Undergoing Radical Cystectomy

**DOI:** 10.3390/biomedicines10040910

**Published:** 2022-04-15

**Authors:** Jae-Hoon Chung, Chung-Un Lee, Dong-Hyeon Lee, Wan Song

**Affiliations:** 1Department of Urology, Samsung Medical Center, Sungkyunkwan University School of Medicine, Seoul 06351, Korea; jaehoontasker.chung@samsung.com (J.-H.C.); iatronices@naver.com (C.-U.L.); 2Department of Urology, Ewha Womans University Medical Center, Ewha Womans University School of Medicine, Seoul 07985, Korea; leedohn@ewha.ac.kr

**Keywords:** micropapillary variant, programmed death ligand-1, recurrence, squamous differentiation, urothelial carcinoma

## Abstract

The expression and prognostic role of programmed death ligand-1 (PD-L1) on tumor-infiltrating immune cells (TICs) has not been determined in urothelial carcinoma (UC) with variant histology. We retrospectively reviewed 90 patients (44 with micropapillary variant of UC (MPUC) and 46 with UC with squamous differentiation (UCSD)) who underwent radical cystectomy between January 2013 and December 2019. The expression of PD-L1 in TICs was measured using the VENTANA (SP-142) immunohistochemistry assay and dichotomized using a 5% cutoff value (positive ≥ 5%). Kaplan–Meier survival analysis was used to estimate recurrence-free survival (RFS), and multivariable Cox proportional hazard models were used to identify factors predicting tumor recurrence. Overall, positive PD-L1 expression in TICs was confirmed in 50 of 90 (55.6%) patients (40.1% (18/44) of MPUC and 69.9% (32/46) of UCSD). RFS was significantly shorter in patients with positive PD-L1 expression in TICs than in those with negative PD-L1 expression both in MPUC (*p* = 0.005) and UCSD (*p* = 0.046). Positive PD-L1 expression in TICs was significantly associated with an increased risk of tumor recurrence in both MPUC (HR = 1.85; 95% CI: 1.323–2.672; *p* = 0.017) and UCSD (HR = 1.58; 95% CI: 1.162–2.780; *p* = 0.032). In conclusion, positive PD-L1 expression in TICs was significantly associated with poorer RFS in both MPUC and UCSD patients. Our results support the use of adjuvant immunotherapy in these patients if they test positive for PD-L1 in their TICs.

## 1. Introduction

Over the past decade, cisplatin-based combination chemotherapy has been the gold standard for neoadjuvant or adjuvant treatment for muscle-invasive urothelial carcinoma (UC) in the bladder [1]. However, platinum-based chemotherapy has inherent nephrotoxicity, which means that approximately half of the patients are not eligible for cisplatin- based chemotherapy [2,3]; thus, they are treated with less effective chemotherapeutic regimens. Recently, immune checkpoint inhibitors (ICIs) have emerged as a promising therapeutic option. Indeed, studies have shown that 20–25% of patients who progressed following chemotherapy to exhibit a durable response with tolerable toxicities [4,5,6]. By blocking the programmed death-1 (PD-1)/programmed death ligand-1 (PD-L1) interaction, T cell-mediated immune response is restored and resistance of cancer cells to immune response can be overcome [7,8]. PD-L1 expression determined by immunohistochemistry (IHC) provides a practical assay and is widely used to help predict treatment response and guide therapeutic decision making [3,9].

In contrast, UC is one of the most histopathologically heterogeneous cancers in which UC with variant histology (UCV) is identified in up to one-third of advanced UC cases [10]. Patients with UCV usually show a poorer response to conventional treatment and a worse clinical course than those with pure UC [11,12]. In addition, the presence of UCV can influence PD-L1 expression and impact prognosis [13]. However, few studies have explored PD-L1 expression in UCV. The current criteria for evaluating PD-L1 expression have been validated in pure UC, but not in UCV [14]. Furthermore, patients with UCV are excluded from major clinical trials [15,16,17], thus limiting the clinical usefulness of ICI in these patients.

The examination of PD-L1 expression in UCV could provide clinical evidence for ICI in these patients. Therefore, in this study, we investigated the expression and prognostic role of PD-L1 on tumor-infiltrating immune cells (TICs) in patients with UCV following radical cystectomy (RC) as a predictive biomarker by analyzing the correlation with tumor recurrence.

## 2. Materials and Methods

### 2.1. Study Population

This study was approved by the Institutional Review Board of Ewha Womans University Mokdong Hospital (IRB No. 2019-04-019). Written informed consent was waived due to the retrospective study design. All study protocols were performed in accordance with the Declaration of Helsinki, and patient data complied with the relevant privacy regulations and data protection.

We retrospectively reviewed a prospectively maintained cystectomy database. In the entire cohort, we identified 731 patients who underwent RC for UC in the bladder between January 2013 and December 2019. From the cohort, we excluded 596 patients who were pathologically diagnosed with pure UC following RC. In addition, the following variant histologies were excluded: glandular differentiation (*n* = 7), plasmacytoid variant (*n* = 11), sarcomatoid differentiation (*n* = 8), neuroendocrine differentiation (*n* = 6), and other rare mixed variants, such as pleomorphic giant cell, syncytiotrophoblastic giant cell, microcytic variants, and signet ring cell adenocarcinoma (*n* = 13). Finally, a total of 90 patients, including 44 with micropapillary variant of UC (MPUC) and 46 with UC with squamous differentiation (UCSD), were analyzed in this study (Figure 1). All patients were preoperatively staged as cM0.

### 2.2. Data Collection

Demographic and pathological data of patients were retrieved from the medical records, including age at surgery, sex, pathologic T and N stages, presence of concomitant carcinoma in situ (CIS), lymphovascular invasion (LVI), number of resected lymph nodes (LNs), surgical margin status, perioperative chemotherapy treatment, and type of urinary diversion.

Tumor recurrence was defined as recurrence at the surgical bed or regional LNs locally and/or distant metastasis in patients previously classified as disease-free. Recurrence-free survival (RFS) was measured from the date of RC to the date of the first documented recurrence or the date of the last follow-up when patients had not yet experienced tumor recurrence.

### 2.3. Histologic Assessment

All surgical procedures were performed as an open approach; RC included the removal of prostate and seminal vesicles in men and removal of both ovaries and uterus in women. Bilateral pelvic LN dissection with a standard template was also performed. Formalin-fixed, paraffin-embedded (FFPE) tissue sections were obtained and stained with hematoxylin and eosin (HE) to identify the presence of tumors. All specimens were reviewed by an experienced pathologist specializing in genitourinary cancer, and the features of MPUC and UCSD were identified. Pathologic staging was determined according to the 2010 TNM classification of the American Joint Committee on Cancer (AJCC), and grading was based on the 2004 World Health Organization (WHO)/International Society of Urologic Pathology consensus classification.

### 2.4. IHC Assay and Scoring

IHC for PD-L1 was performed using 4 μm-thick tissue sections. All staining was performed using a Ventana BenchMark Ultra System (Ventana Medical Systems, Tucson, AZ, USA), and the OptiView DAB IHC Detection Kit (Ventana Medical Systems) was used to visualize the antigen–antibody reaction according to the manufacturer’s protocol. Specimens were stained with the PD-L1 IHC clone SP142 (Ventana Medical System; retrieval: CC1 48 min; incubation: 16 min; ready to use dilution).

All slides were reviewed by an experienced pathologist blinded to the clinicopathological and recurrence data of the patients. PD-L1 expression in TICs was assessed as the percentage of PD-L1 expressing leukocytes and/or macrophages within the tumor area. PD-L1 expression was dichotomized as positive (≥5%) or negative (<5%) using a 5% cutoff value for statistical analysis. Representative images are shown in Figure 2.

### 2.5. Follow-up Protocol

After RC, each patient had a regular follow-up schedule according to the guidelines [18] and institutional protocols. Briefly, patients were scheduled at one month postoperatively to check for physical examination, laboratory tests, and diet. Afterward, they were scheduled to visit every three months for the first two years and then every six months for the next three years. At every visit, laboratory tests, urine analysis with cytology, computed tomography (CT), or magnetic resonance imaging (MRI) of the chest, abdomen, and pelvis were conducted to check for tumor recurrence. Bone scintigraphy was performed when clinically indicated.

### 2.6. Statistical Analysis

Descriptive statistics (mean, standard deviation (SD), median, and interquartile range (IQR)) were used for continuous variables, and categorical variables were presented as absolute values (percentages). An independent *t*-test was used to compare continuous variables, and Pearson’s chi-square test or Fisher’s exact test was used to compare categorical variables. Survival analysis was performed using both the Kaplan–Meier and Cox regression analyses. Kaplan–Meier survival analysis was used to estimate RFS in each group in relation to PD-L1 expression, and differences were assessed using the log-rank test. Multivariable Cox proportional hazard models were used to analyze the impact of PD-L1 expression on RFS after adjusting for available confounders. All statistical analyses were performed using the IBM SPSS Statistics for Windows, version 23.0 (IBM Corp. Armonk, NY, USA). Statistical significance was set at *p* < 0.05.

## 3. Results

### 3.1. Demographic and Tumor Characteristics

Table 1 shows the baseline clinicopathological characteristics of the 90 patients who underwent RC for MPUC (*n* = 44) or UCSD (*n* = 46). The median (IQR) age at RC was 68.0 (61.8–73.0) years, and most patients were men (*n* = 77, 85.6%) with 14.4% (*n* = 13) being women. Muscle invasive UC was identified in 82.2% (*n* = 74) of patients, of whom 73.0% (54/74) had locally advanced tumor stage ≥ pT3. The median (IQR) resected LN count was 20.0 (14.0–29.0), and positive LN involvement was documented in 43 (47.8%) patients. LVI invasion was identified in 43 (47.8%) patients, and a positive surgical margin was found in 14 (15.6%) patients. After RC, urinary diversion with orthotopic neobladder was performed in 76 (84.6%) patients. In total, 15 (16.7%) and 60 (66.7%) patients received neoadjuvant and adjuvant chemotherapy, respectively. When patients were categorized according to the type of histologic variant, the pathologic tumor stage was found to be more advanced in patients with UCSD than in those with MPUC (*p* = 0.013). Patients with UCSD received more neoadjuvant chemotherapy than those with MPUC (*p* = 0.022). There was no difference between the two groups in terms of other clinicopathological characteristics.

### 3.2. Relationship between PD-L1 Expression and Clinicopathological Features

Table 2 summarizes the clinical and pathological characteristics of patients stratified according to PD-L1 expression in TICs. Overall, positive PD-L1 expression was confirmed in 50 of 90 (55.6%) patients with MPUC or UCSD. PD-L1 expression was positive in 40.1% (18/44) of patients with MPUC. Moreover, positive PD-L1 expression was significantly associated with positive LN involvement (*p* = 0.030). For patients with UCSD, the rate of positive PD-L1 expression was 69.9% (32/46), which was higher than that in patients with MPUC (*p* = 0.006). In these patients, positive PD-L1 expression was significantly different according to tumor stage (*p* = 0.005). However, other clinicopathological factors were not significantly related to PD-L1 expression in TICs.

### 3.3. Relationship between PD-L1 Expression and RFS

Follow-up data were available for all patients, and during the median (IQR) follow-up of 30.4 (18.2–39.8) months, local recurrence and/or distant metastasis were documented in 52.2% (47/90) of patients. The overall RFS rates according to PD-L1 expression on TICs were estimated using the Kaplan–Meier method as shown in Figure 3. For patients with MPUC, the two-year overall RFS rate was significantly (*p* = 0.005) shorter in patients with positive PD-L1 expression (12.0%) than in patients with negative PD-L1 expression (63.4%), as seen in Figure 3A. For patients with UCSD, PD-L1 expression in TICs also significantly (*p* = 0.046) affected the two-year overall RFS rate, which was 78.6% in patients negative for PD-L1 and 42.6% in those positive for PD-L1 (Figure 3B).

Table 3 summarizes the multivariable Cox proportional hazard regression analysis for predicting tumor recurrence after RC. A locally advanced tumor stage more than ≥ pT3, LN involvement and LVI (only for patients with MPUC, all *p* < 0.05), and positive PD-L1 in TICs were also significantly associated with an increased risk of tumor recurrence both in patients with MPUC (hazard ratio (HR) = 1.85; 95% confidence interval (CI): 1.323–2.672; *p* = 0.017), and in patients with MPUC (HR = 1.58; 95% CI: 1.162–2.780; *p* = 0.032).

## 4. Discussion

In this study, we examined the expression of PD-L1 in TICs of patients with MPUC or UCSD following RC and identified the prognostic implication of PD-L1 in these patients. Positive PD-L1 expression in TICs was confirmed in 40.1% (18/44) of patients with MPUC and 69.9% (32/46) of patients with UCSD. We found that positive PD-L1 expression was associated with adverse pathologic characteristics (advanced T stage or positive LN). In addition, we observed a significant difference in tumor recurrence following RC depending on PD-L1 expression in the TICs in these patients, and the difference in tumor recurrence depending on PD-L1 expression was evident on multivariable analysis. These results suggest that adjuvant immunotherapy should be considered for patients with MPUC or UCSD who test positive for PD-L1 in TICs following RC beyond simply adverse pathologic criteria. To the best of our knowledge, this is the first and largest study to evaluate the expression and prognostic implications of PD-L1 on TICs in patients with MPUC or UCSD.

In general, positive PD-L1 expression was identified in approximately 15–35% of pure UC obtained from post-cystectomy specimens, and the PD-L1 positivity rate was higher in UCV than in pure UC [3,19,20]. Previous studies have reported that high levels of PD-L1 expression are associated with aggressive and/or advanced tumors in the bladder [9,21]. Therefore, in our study, positive PD-L1 expression in TICs was reported in 40.1% (18/44) and 69.9% (32/46) of MPUC and UCSD patients, respectively. These results were consistent with those obtained in a study by Reis et al. [3], in which IC2/3 (high PD-L1 expression) levels were 31.6% (6/19) of MPUC and 75.0% (12/16) of UCSD using atezolizumab criteria. Although, it has also been shown that when positive PD-L1 expression was assessed using the combined positive score (CPS ≥10) defined as the number of PD-L1 staining cells (tumor cells, macrophages, and lymphocytes) divided by the total number of viable tumor cells and multiplied by 100, PD-L1 positivity was found in 40.0% (2/5) of MPUC and in 100.0% (5/5) of UCSD [22]. Collectively, there appears to be a vital difference between PD-L1 expression in pure UC and UCV.

In our study, concerning the clinicopathological features, PD-L1 expression did not appear to be affected by age or sex, which was consistent with the results of previous studies [22,23]. However, positive PD-L1 expression in TICs was significantly associated with positive LN involvement in patients with MPUC (*p* = 0.030) and tumor stage in patients with MPUC (*p* = 0.006). Wang et al. [24] showed that in pure UC, positive PD-L1 expression in TICs was positively associated with aggressive pathologic features, including tumor size, tumor stage, nodal status, and histologic grade (all *p* < 0.005). Together, these findings suggest that positive PD-L1 expression on TICs is associated with adverse pathologic findings in both pure UC and UCV.

In the tumor microenvironment, PD-L1 plays a crucial role in maintaining a balance between immune inhibition and activation [25,26]. Previous studies have reported the development of an exhausted immune state in the tumor stroma of UC [24], and positive PD-L1 expression in TICs showing a significantly shorter RFS compared to negative PD-L1 expression [24,27]. Therefore, we previously reported that positive PD-L1 expression in TICs was significantly associated with shorter RFS in “high-risk” (≥pT3a and/or pN+) UC and may be used as a prognostic biomarker of tumor recurrence following RC irrespective of adjuvant chemotherapy [28,29]. Additionally, in our study, we found that positive PD-L1 expression in TICs was significantly associated with an increased risk of tumor recurrence in patients with MPUC or UCSD. Thus, these results expand the prognostic value of PD-L1 expression in TICs in both pure UC and MPUC or UCSD.

In general, clinical outcome of UCV is more aggressive than that of pure UC. In a previous study [28], a total of 219 patients with “high-risk” (≥pT3a and/or pN+) pure UC, patients with ≥pT3a and pN+ were identified in 86.7% (190/219) and 53.4% (117/102). In a current study, a total of 90 patients with UCV, patients with ≥pT3a and pN+ were identified in 60.0% (54/90) and 47.8% (43/90). However, during the median follow-up of 32.5 months and 30.4 months, local recurrence and/or distant metastasis were identified in 115 (52.5%) and 47 (52.2%) patients, respectively. Therefore, these results suggest that adjuvant immunotherapy should be considered for patients with MPUC or UCSD who test positive for PD-L1 in TICs following RC

Recently, two phase III trials evaluated the role of adjuvant ICI in high-risk UC in the bladder but reported inconsistent results. In the IMvigor101 study, the trial did not show the improvement of disease-free survival (DFS) in the atzolizumab group over observation (19.4 months vs. 16.6 months, HR = 0.89; 95% CI: 0.74–1.08; *p* = 0.24) [16]. However, in the CheckMate-274 study, there was a significant improvement of DFS in the nivolumab group than in the placebo group (20.8 months vs. 10.8 months) [15]. The disease-free rate at six months was 74.9% with nivolumab and 60.3% with the placebo in the intention-to-treatment population (HR 0.70; 98.22% CI: 0.55–0.90; *p* < 0.001), and 74.5% with nivolumab and 55.7% with the placebo in patients with PD-L1 expression of more than 1% (HR 0.55; 98.72% CI: 0.35–0.85; *p* < 0.001) [15]. Therefore, ongoing phase III clinical trials (AMBASSADOR trial, NCT3244384, adjuvant pembrolizumab vs. observation) may provide more insights into treatment strategies. Collectively, additional trials are required to identify the clinical role of adjuvant ICI, and further studies on patients with UCV should be conducted to confirm the role of positive PD-L1 on TICs as prognostic biomarkers.

Despite the clinical implications of our study, there are several limitations. For instance, its non-randomized, retrospective design and relatively small number of cases may raise concerns of selection bias. However, as UCV is not common, our study included as many cases as possible and reflected real-world clinical practice. In addition, given the multiple PD-L1 assays, we used the VENTANA assay with the SP 142 antibody because, in the past, atezolizumab was only reimbursed based on the results of the VENTANA test. Therefore, in interpreting the results, differences due to interpretation scores, quantification methods, and inter-assay discordance should be considered. Finally, as no patients have received immunotherapy, the current study cannot reach a definite decision for the prediction of the most representative diagnostic assay and therapeutic response. Therefore, a large, prospective, randomized study is required to validate the results reported here.

## 5. Conclusions

In summary, positive PD-L1 expression in TICs is associated with higher tumor stage or LN involvement in patients with MPUC or UCSD, and a worse oncologic outcome in terms of tumor recurrence. These results may help identify patients who are at risk of tumor recurrence and support the use of adjuvant immunotherapy in PD-L1-positive patients with MPUC or UCSD. In future clinical trials, inclusion of UCV is needed to verify whether positive PD-L1 expression in TICs is associated with a more favorable response to immunotherapy.

## Figures and Tables

**Figure 1 biomedicines-10-00910-f001:**
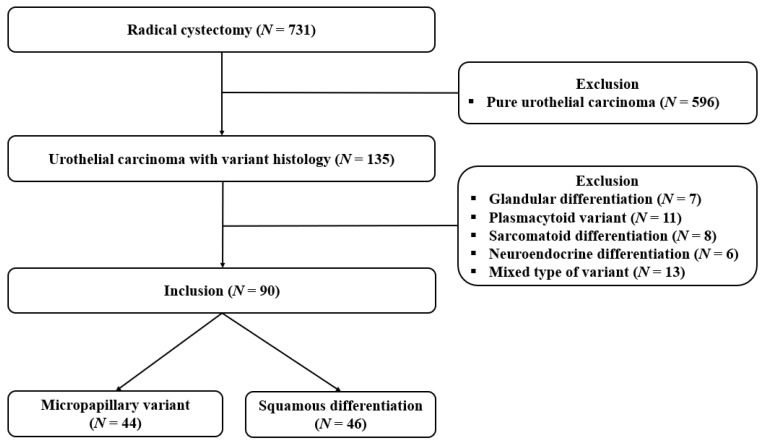
Flow chart of study inclusion.

**Figure 2 biomedicines-10-00910-f002:**
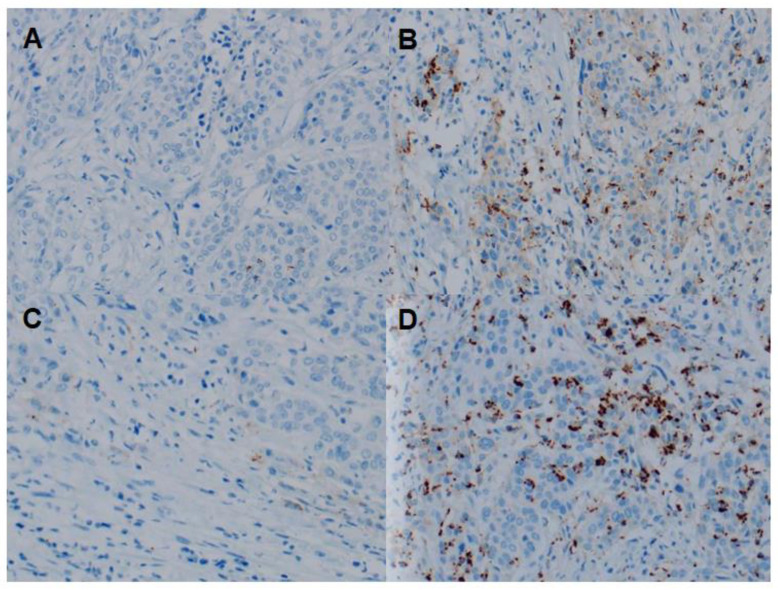
Representative images of PD-L1 expression in TICs using the VENTANA (SP142) immunohistochemistry assay. (**A**) Negative expression of PD-L1 in MPUC, (**B**) Positive expression of PD-L1 in MPUC. (**C**) Negative expression of PD-L1 in UCSD, (**D**) Positive expression of PD-L1 in UCSD. All images are ×200 magnification. PD-L1, programmed death-ligand 1; TICs, tumor-infiltrating immune cells; MPUC, micropapillary variant of urothelial carcinoma; UCSD, urothelial carcinoma with squamous differentiation.

**Figure 3 biomedicines-10-00910-f003:**
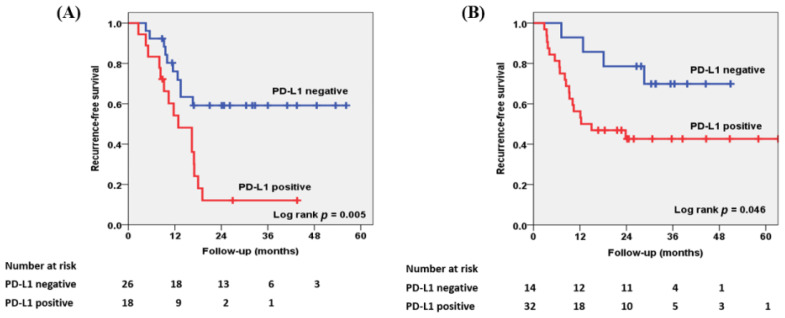
Kaplan–Meier survival curves for RFS according to PD-L1 expression in TICs in patients with (**A**) MPUC and (**B**) UCSD. For patients with MPUC, the two-year overall RFS rate was significantly (*p* = 0.005) shorter in patients with positive expression of PD-L1 in TICs than in those with negative expression of PD-L1 in TICs (12.0% and 63.4%, respectively). Similarly, for patients with UCSD, PD-L1 expression in TICs also significantly (*p* = 0.046) affected the two-year overall RFS rate, which was 78.6% in patients with negative PD-L1 expression in TICs and 42.6% in PD-L1-positive patients.

**Table 1 biomedicines-10-00910-t001:** Baseline characteristics of 90 patients with urothelial carcinoma with variant histology.

Parameters	Total	Variant Histology		*p*
		Micropapillary	Squamous Differentiation	
No. of patients	90 (100.0)	44 (48.9)	46 (51.1)	
Age at surgery, years				0.529
Median (IQR)	68.0 (61.8–73.0)	68.5 (62.3–70.8)	66.5 (58.3–74.3)	
Mean (SD)	66.0 (10.1)	66.7 (7.9)	65.3 (11.8)	
Sex, n (%)				0.831
Male	77 (85.6)	38 (86.4)	39 (84.8)	
Female	13 (14.4)	6 (13.6)	7 (15.2)	
Pathologic T stage at RC, n (%)				0.013
≤pT1	16 (17.8)	13 (29.5)	3 (6.5)	
pT2	20 (22.2)	7 (16.0)	13 (28.3)	
pT3/4	54 (60.0)	24 (54.5)	30 (65.2)	
Concomitant CIS at RC, n (%)				0.057
Yes	46 (51.1)	27 (61.4)	19 (41.3)	
No	44 (48.8)	17 (38.6)	27 (58.7)	
LVI at RC, n (%)				0.090
Yes	47 (52.2)	27 (61.4)	20 (43.5)	
No	43 (47.8)	17 (38.6)	26 (56.5)	
No. of resected LNs at RC				0.112
Median (IQR)	20.0 (14.0–29.0)	21.0 (14.3–30.5)	18.5 (13.0–25.3)	
Mean (SD)	22.5 (10.9)	24.4 (11.8)	20.8 (9.7)	
Pathologic N status at RC, n (%)				0.093
Negative	47 (52.2)	19 (43.2)	28 (60.9)	
Positive	43 (47.8)	25 (56.8)	18 (39.1)	
Surgical margin status, n (%)				0.210
Negative	76 (84.4)	35 (79.5)	41 (89.1)	
Positive	14 (15.6)	9 (20.5)	5 (10.9)	
Type of urinary diversion, n (%)				0.283
Ileal conduit	14 (15.6)	5 (11.4)	9 (19.6)	
Orthotopic neobladder	76 (84.8)	39 (88.6)	37 (80.4)	
Neoadjuvant chemotherapy, n (%)				0.022
Yes	15 (16.7)	3 (6.8)	12 (26.1)	
No	75 (83.3)	41 (93.2)	34 (73.9)	
Adjuvant chemotherapy, n (%)				0.766
Yes	60 (66.7)	30 (68.2)	30 (65.2)	
No	30 (33.3)	14 (31.8)	16 (34.8)	

IQR, interquartile range; SD, standard deviation; RC, radical cystectomy; CIS, carcinoma in situ; LVI, lymphovascular invasion; LN, lymph node.

**Table 2 biomedicines-10-00910-t002:** Association of PD-L1 expression and clinicopathologic characteristics.

Parameters	Variant Histologies							
	Micropapillary				Squamous Differentiation			
	Total	PD-L1 Score on TICs		*p*	Total	PD-L1 Score on TICs		*p*
		Negative	Positive			Negative	Positive	
N	44 (100.0)	26 (59.1)	18 (40.1)		46 (100.0)	14 (30.4)	32 (69.6)	
Age				0.888				0.371
<68.0	19 (43.2)	11 (57.9)	8 (42.1)		25 (54.3)	9 (36.0)	16 (60.0)	
≥68.0	25 (56.8)	15 (60.0)	10 (40.0)		21 (45.7)	5 (23.8)	16 (72.2)	
Sex				0.375				1.000
Male	38 (86.4)	21 (55.3)	17 (44.7)		39 (84.8)	12 (30.8)	27 (69.2)	
Female	6 (13.6)	5 (83.3)	1 (16.7)		7 (15.2)	2 (28.6)	5 (71.4)	
Tumor stage				0.467				0.005
≤pT2	20 (45.5)	13 (65.0)	7 (35.0)		16 (34.8)	9 (56.2)	7 (43.8)	
≥pT3	24 (54.5)	13 (54.2)	11 (45.8)		30 (65.2)	5 (16.7)	25 (83.3)	
Concomitant CIS				0.510				0.611
Yes	27 (61.4)	17 (63.0)	10 (37.0)		19 (41.3)	5 (26.3)	14 (73.7)	
No	17 (38.6)	9 (52.9)	8 (47.1)		27 (58.7)	9 (33.3)	18 (66.7)	
LVI				0.977				0.555
Yes	27 (61.4)	16 (59.3)	11 (40.7)		20 (43.5)	7 (35.0)	13 (65.0)	
No	17 (38.6)	10 (58.8)	7 (41.2)		26 (56.5)	7 (26.9)	19 (73.1)	
Lymph node positivity				0.030				0.188
Negative	19 (43.2)	15 (78.9)	4 (21.1)		28 (60.9)	11 (39.3)	17 (60.7)	
Positive	25 (56.8)	11 (44.0)	14 (56.0)		18 (39.1)	3 (16.7)	15 (83.3)	

CIS, carcinoma in situ; LVI, lymphovascular invasion; PD-L1, programmed death-ligand 1; TIC, tumor-infiltrating immune cell.

**Table 3 biomedicines-10-00910-t003:** Multivariable Cox proportional hazard regression analyses to predict tumor recurrence.

Variables	Micropapillary			Squamous Differentiation		
	HR	95% CI	*p*	HR	95% CI	*p*
Age						
<68.0	ref.			ref.		
≥68.0	1.01	0.992–1.043	0.151	1.04	0.963–1.082	0.260
Sex						
Male	ref.			ref.		
Female	1.23	0.535–2.231	0.532	1.44	0.441–2.753	0.762
Tumor stage						
≤pT2	ref.			ref.		
≥pT3	2.24	1.430–4.262	0.022	2.03	1.421–3.757	0.010
Concomitant CIS						
No	ref.			ref.		
Yes	1.30	0.692–2.433	0.410	1.17	0.716–1.632	0.343
LVI						
No	ref.			ref.		
Yes	1.65	1.240–2.530	0.038	1.44	0.993–2.046	0.062
Lymph node positivity						
Negative	ref.			ref.		
Positive	3.02	1.782–5.428	0.001	2.49	1.672–3.631	0.004
Surgical margin status						
Negative	ref.			ref.		
Positive	1.32	0.970–2.026	0.092	1.46	0.820–2.382	0.264
PD-L1 in TICs						
Negative	ref.			ref.		
Positive	1.85	1.323–2.672	0.017	1.58	1.162–2.780	0.032

CIS, carcinoma in situ; LVI, lymphovascular invasion; PD-L1, programmed death-ligand 1; TIC, tumor-infiltrating immune cell; HR, hazard ratio; CI, confidence interval.

## Data Availability

The data presented in this study are available on request from the corresponding author. The data are not publicly available due to their containing information that could compromise the privacy of research participants.

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
