# Peer review of "Expression and Prognostic Implication of PD-L1 in Patients with Urothelial Carcinoma with Variant Histology (Squamous Differentiation or Micropapillary) Undergoing Radical Cystectomy"

_biomedicines, 2022, doi:10.3390/biomedicines10040910_

Round 1

Reviewer 1 Report

An interesting paper where histological findings are compared to outcome and discussed around modern UC treatment. Well written and clinical relevant.

An error is seen in the absctract where in row 24 and 25 it´s written MPUC in both places.

In material and methods, page 4 row 109 there is no explantion around the cut of at 5% which could be expanded.

Author Response

 An interesting paper where histological findings are compared to outcome and discussed around modern UC treatment. Well written and clinical relevant.

1. An error is seen in the abstract where in row 24 and 25 it´s written MPUC in both places.

 -->We corrected the mistake. We changed MPUC to UCSD on 1 page, line 25.

2. In material and methods, page 4 row 109 there is no explanation around the cut of at 5% which could be expanded.

--> PD-L1 expression more than 5% in TICs based on the results of VENTANA test using the SP 142 antibody is associated with increased objective response rate to atezolizumab. In addition, in the past, atezolizumab treatment was only reimbursed by the government based on the results of VENTANA test using the SP 142 antibody. Therefore, we used a 5% cutoff value.

Reviewer 2 Report

The authors presented that positive programmed death ligand-1 (PD-L1) expression in tumor-infiltrating immune cells (TICs) by immunostaining was significantly associated with poorer recurrence-free survival (RFS) in both micropapillary variant of urothelial carcinoma (MPUC) and UC with squamous differentiation (UCSC) patients. The results are valuable because they are in rare bladder cancer variants, especially MPUC, but they differ little from the results of the usual type. There are some questions and suggestions as described below.

Major comments

  • The authors should present typical histology of both MPUC and UCSC by photos. They should also present the positive and negative cases of them by photos. They make clear to understand importance and condition of this experiment.
  • How about PD-L1 expressions of UC itself? In this experiment, the authors only focused on PD-L1 expressions of TICs. Differences in PD-L1 expression in UC and TICs should provide interesting data for prognosis and therapy. They should better to discuss the PD-L1 expression difference in UC and TICs in pure UC and UC variants.
  • Because MPUC and UCSC are aggressive variants of UC, the authors should explain the differences in stage and recurrence duration compared to pure UC, and discuss them more with PD-L1 staining in Discussion.

Minor comments

  • What means of “cM” on page 2, line 76?
  • The authors should explain “combined positive score” on page 8, line 217.

Author Response

 The authors presented that positive programmed death ligand-1 (PD-L1) expression in tumor-infiltrating immune cells (TICs) by immunostaining was significantly associated with poorer recurrence-free survival (RFS) in both micropapillary variant of urothelial carcinoma (MPUC) and UC with squamous differentiation (UCSD) patients. The results are valuable because they are in rare bladder cancer variants, especially MPUC, but they differ little from the results of the usual type. There are some questions and suggestions as described below.

Major comments

1. The authors should present typical histology of both MPUC and UCSD by photos. They should also present the positive and negative cases of them by photos. They make clear to understand importance and condition of this experiment.

--> As your recommendation, we added figures showing the typical histology of MPUC, UCSD and each PD-L1 expression. Please see Figure 2 in revised manuscripts on page 4.

2. How about PD-L1 expressions of UC itself? In this experiment, the authors only focused on PD-L1 expressions of TICs. Differences in PD-L1 expression in UC and TICs should provide interesting data for prognosis and therapy. They should better to discuss the PD-L1 expression difference in UC and TICs in pure UC and UC variants.

--> As you mentioned, comparison of PD-L1 expression of UC in pure UC and UC variant could provide an important information in prognosis and treatment plan. However, as VENTANA test using the SP 142 antibody was optimized for the detection of PD-L1 expression in TICs, we could not measure the expression of PD-L1 in UC. As your recommendation, we will report the difference of PD-L1 expression both in UC and TICs in pure UC and UC variants in a follow-up study.

 In addition, on page 8 line 215-228, we discussed the difference between PD-L1 expression in TICs in pure UC and UC variants. In general, PD-L1 positivity rate was higher in UC variants than in pure UC. Previous studies reported that positive PD-L1 expression in TICs was reported in 15%-35% of pure UC. However, in our study, positive PD-L1 expression in TICs was reported in 40.1% (18/44) and 69.9% (32/46) of MPUC and UCSD patients, respectively. Possible explanation is the higher mutational burden in UC variants compared to pure UC. Collectively, there appears to be a vital difference between PD-L1 expression in Pure UC and UC variants.

3. Because MPUC and UCSD are aggressive variants of UC, the authors should explain the differences in stage and recurrence duration compared to pure UC, and discuss them more with PD-L1 staining in Discussion.

 --> As your recommendation, we added the difference and clinical course between pure UC and UC variants. Please see the revised manuscript page 9, line 249-257 starting with the sentence as “In general, clinical outcome of UCV is more aggressive…………….”

Minor comments

1. What means of “cM” on page 2, line 76?

--> We corrected the mistake. We modified cM to cM0 in line 76.

2. The authors should explain “combined positive score” on page 8, line 217.

--> Combined positive score is defined as the number of PD-L1 staining cells (tumor cells, lymphocytes and macrophage) divided by the total number of viable tumor cells and multiplied by 100. Please see the revised manuscript page 8, line 224-226 starting with “defined as the number of…… and multiplied by 100”.

Round 2

Reviewer 2 Report

Authors ameliorated the manuscript with reviewer’s comments. It becomes acceptable for the journal.